# Hierarchical conductive metal-organic framework films enabling efficient interfacial mass transfer

Chuanhui Huang [1,11], Xinglong Shang[2,11], Xinyuan Zhou[3,11], Zhe Zhang[1,4], Xing Huang[1], Yang Lu[1], Mingchao Wang [1], Markus Löffler[5], Zhongquan Liao[6], Haoyuan Qi[1,7], Ute Kaiser [7], Dana Schwarz[8], Andreas Fery [1,8], Tie Wang[3], Stefan C. B. Mannsfeld [4], Guoqing Hu [2] ✉, Xinliang Feng [1,9] ✉ & Renhao Dong [1,10] ✉

Heterogeneous reactions associated with porous solid films are ubiquitous and play an important role in both nature and industrial processes. However, due to the no-slip boundary condition in pressure-driven flows, the interfacial mass transfer between the porous solid surface and the environment is largely limited to slow molecular diffusion, which severely hinders the enhancement of heterogeneous reaction kinetics. Herein, we report a hierarchical-structure-accelerated interfacial dynamic strategy to improve interfacial gas transfer on hierarchical conductive metal-organic framework (*c*-MOF) films. Hierarchical *c*-MOF films are synthesized via the in-situ transformation of insulating MOF film precursors using π-conjugated ligands and comprise both a nanoporous shell and hollow inner voids. The introduction of hollow structures in the *c*-MOF films enables an increase of gas permeability, thus enhancing the motion velocity of gas molecules toward the *c*-MOF film surface, which is more than 8.0-fold higher than that of bulk-type film. The *c*-MOF film-based chemiresistive sensor exhibits a faster response towards ammonia than other reported chemiresistive ammonia sensors at room temperature and a response speed 10 times faster than that of the bulk-type film.

Porous solids, ranging from classic inorganic zeolites to molecular organic and hybrid frameworks[1–3], have fascinating properties for various heterogeneous reactions, such as electrochemical energy storage[4–6], heterogeneous catalysis[7–10], and sensing[11–14]. Regarding heterogeneous reactions involving porous solid films, mass transfer at solid/liquid or solid/gas interfaces plays an important role in reaction kinetics, and the overall rate of reaction is usually limited by the interfacial mass transfer[15–17]. However, subject to the no-slip boundary condition in pressure-driven flows[18,19], mass transfer near the surface of the solid film is largely limited to the relatively slow process of molecular diffusion since the convective movement of fluids approaches zero (Fig. 1a). The critical bottleneck to promote convective movement

at the boundary layer results from high hydraulic resistance, i.e., the low permeability of solid film samples[20,21]. Such a low permeability induced stagnant boundary layer poses a grand challenge for further promotion of interfacial mass transfer, necessitating the design of innovative porous nanointerfaces to boost interfacial mass transfer.

Conductive metal–organic frameworks (*c*-MOFs), which are electroactive crystalline porous coordination polymers[22–24], are emerging as promising electronic materials and have exhibited potential for broad applications such as electronic devices[25], electrocatalysis[26,27] and energy storage[28,29]. Due to their high porosity, intrinsic electrical conductivity and abundant functional sites, *c*-MOF films have been widely utilized as active layers for chemiresistive gas sensing[30–32]. In a *c*-MOF-

**Fig. 1 | Schematic of interfacial mass transport on a solid porous film. a** Flow velocity vectors and slow mass transfer (concentration gradient-induced molecular diffusion) on the surface of a solid porous film (left) and the corresponding concentration field on the surface of the solid porous film (right). **b** Flow velocity vectors and fast mass transfer (molecular diffusion plus convection) on the surface of a hierarchical porous film (left) and the corresponding concentration field on the surface of the hierarchical porous film (right). The black arrow indicates the flow direction, and the length of the arrow indicates the magnitude of the velocity. The red spots represent the molecules that reach the surface of the film.

based chemiresistive gas sensing system, the sensing relies heavily on the adsorption of target gas molecules on the surface of c-MOF films, and the kinetics of this kind of heterogeneous reaction are mainly influenced by interfacial mass transfer[33,34]. Therefore, the c-MOF-based chemiresistive gas sensing can be considered a model for establishing the permeability-interfacial mass transfer correlation, providing guidelines to address the challenge of accelerating the interfacial mass transfer for porous solid systems.

Herein, we demonstrate a hierarchical-structure-accelerated interfacial dynamic (HSAID) strategy for the promotion of interfacial mass transfer to boost heterogeneous reactions. Hierarchical c-MOF films (denoted Zn-HHTP-H, PcCu-Zn-H and Co-HHTP-H) were constructed by an in-situ transformation of three-dimensional (3D) ZIF-8 ($Zn(MeIM)_2$, where MeIM = 2-methylimidazole) or ZIF-67($Co(MeIM)_2$) film precursors using π-conjugated ligands (2,3,6,7,10,11-hexahydroxytriphenylene (HHTP) or 2,3,9,10,16,17,23,24-octahydroxyphthalocyaninato copper (PcCu-$(OH)_8$) on a solid surface. The resultant crystalline c-MOF films possess hierarchical structures with a nanoporous shell (~1.2 nm) and hollow interior voids (~500 nm). Systematic gas permeability tests and computational fluid dynamics (CFD) simulations revealed that the permeability of hierarchical hollow Zn-HHTP-H films was 8.4-fold greater than that of a bulk-type Zn-HHTP film, greatly increasing the interfacial mass transfer rate (Fig. 1b). The prepared c-MOF films were further integrated as chemiresistors for ammonia sensing to elucidate the influence of their hollow nature on interfacial mass transfer. Compared to the bulk-type films (MOF films without hierarchical porous structures synthesized by the hydrothermal method), the hierarchical c-MOF film-based sensors displayed a 10-fold improvement in the response speed of ammonia. In particular, the Zn-HHTP-H-based chemiresistive ammonia sensor exhibited the fastest response speed (response time of 9.1 s) at room temperature, superior to those of previously reported chemiresistive ammonia sensors (with response time ≥35 s)[32,33,35]. Our work provides a general synthetic strategy for constructing hierarchical porous structures, thus improving the interfacial mass transfer to accelerate heterogeneous reactions, which can be further employed to achieve high-performance devices.

## Results

### Synthesis and characterization of hierarchical c-MOF films

Hierarchical c-MOF films were constructed on a silicon substrate based on an accessible insulating MOF-to-c-MOF transformation approach (Fig. 2a). In the first step, a typical ~500 nm thick ZIF-8 film was synthesized on silicon substrate[36], which was utilized as the sacrificial precursor (Supplementary Fig. 1). Second, upon immersion into an HHTP solution (ethanol:water = 7:1, v/v) at room temperature, the as-prepared ZIF-8 film was in-situ transformed into a hierarchical c-MOF (Zn-HHTP-H) film (Fig. 2b, e). Through this sacrificial-template synthetic method, the white ZIF-8 films were gradually decomposed while dark purple Zn-HHTP-H films were formed after 24 h (Supplementary Figs. 2, 3). DFT calculations revealed that the transformation reaction from ZIF-8 crystals to Zn-HHTP crystals is thermodynamically spontaneous (Supplementary Table 1). This should be attributed to the greater stability of the square planar linkages of $ZnO_4$ in Zn-HHTP than the Zn-N coordination bonds in ZIF-8 (Supplementary Fig. 4)[37]. Following the same sacrificial template synthetic method, the PcCu-Zn and Co-HHTP films with hierarchical hollow nanostructures were synthesized (named PcCu-Zn-H and Co-HHTP-H, respectively, Fig. 2c–g, Supplementary Figs. 5–8). All these hierarchical c-MOF films exhibited intrinsic electrical conductivity, while the inside was hollow (Fig. 2f, g, Supplementary Table 2).

As shown in the transmission electron microscope (TEM) image (Fig. 3a and Supplementary Fig. 9), the thickness of the Zn-HHTP-H hollow shell was approximately 20 nm, suggesting that the Zn-HHTP-H film possessed a very thin upper surface. The selected-area electron diffraction (SAED) pattern verified the polycrystalline feature of the hollow Zn-HHTP film (Fig. 3b). The lattice fringes at interplanar spacings of 1.8 nm and 0.31 nm corresponded to the (100) and (022) planes of Zn-HHTP crystals, respectively (Fig. 3c)[38]. The honeycomb pattern observed by high-resolution transmission electron microscopy (HRTEM) illustrated the highly ordered hexagonal MOF frameworks along c-axis (Fig. 3d). Elemental mappings by electron energy loss spectroscopy (EELS) confirmed the homogeneous distribution of C, O, and Zn throughout Zn-HHTP-H (Supplementary Fig. 9).

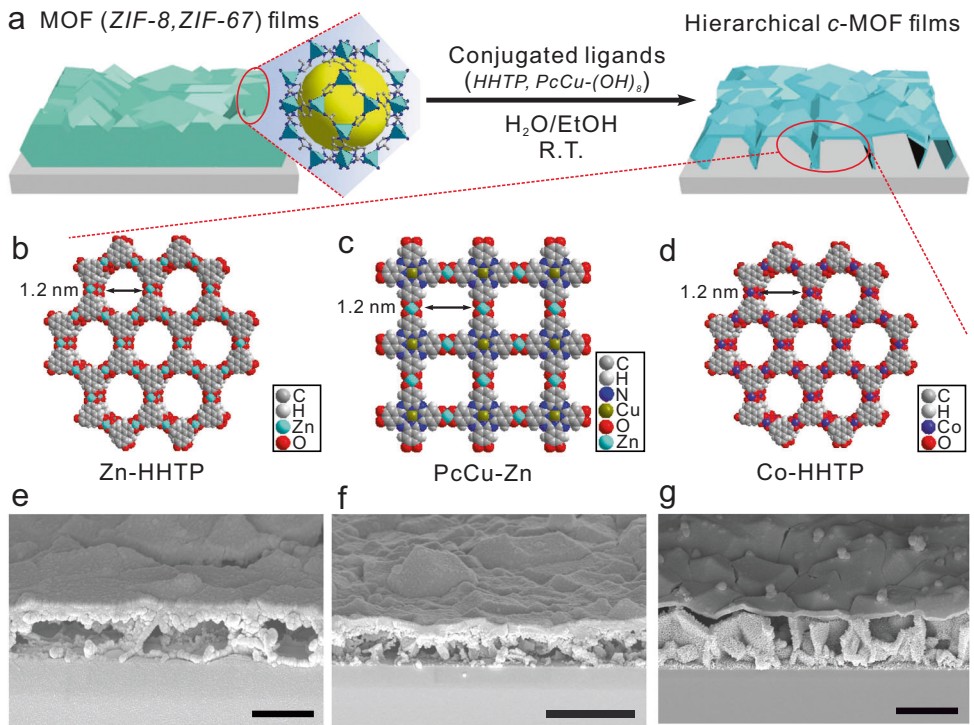

**Fig. 2 | Synthetic strategy and structural characterization. a** Schematic overview of the transformation of insulating 3D MOF film precursors to hierarchical *c*-MOF films. **b**–**d** Schematic structures of Zn-HHTP, PcCu-Zn, and Co-HHTP, respectively. **e**–**g** The corresponding cross-sectional scanning electron microscope (SEM) images of the Zn-HHTP-H, PcCu-Zn-H, and Co-HHTP-H films, respectively. The scale bars represent 500 nm for (**e**), 1 μm for (**f**), and 500 nm for (**g**).

## Morphology control of *c*-MOF films

We suggest that the transformation followed a "dissolution-recrystallization" mechanism[39,40], and the following possible reaction pathway was proposed (Supplementary Figs. 10–13a):

$$2HHTP \rightleftharpoons 2(HHTP^*)^{3-} + 6H^+ \qquad (1)$$

$$3Zn(MeIM)_2 + 6H^+ \rightleftharpoons 6HMeIM + 3Zn^{2+} \qquad (2)$$

$$3Zn^{2+} + 2(HHTP^*)^{3-} \rightleftharpoons Zn_3(HHTP^*)_2 \qquad (3)$$

The total reaction is represented as follows:

$$3Zn(MeIM)_2 + 2HHTP = Zn_3(HHTP^*)_2 + 6HMeIM \qquad (4)$$

A proper balance between the etching and coordination rates was found to be pivotal for achieving a well-defined hollow nanostructure (Supplementary Figs. 13b–19). Herein, the ethanol was introduced to reduce the water-dependent etch rate. The Zn-HHTP-H film with ~500 nm cavities was obtained only at low water fractions (25%) and low temperatures (40 °C) (Supplementary Figs. 13b, S13d and S15). When the water fraction and temperature were increased (e.g., 37.5% and 40 °C, respectively), a Zn-HHTP film was formed with small cavities of ~175 nm (denoted as Zn-HHTP-HS film, Supplementary Figs. 13b, S13e and S16, 17). For comparison, a bulk-type Zn-HHTP film (denoted Zn-HHTP-B) with a thickness of ~500 nm was also synthesized directly on a silicon wafer via a procedure similar to that described previously (Supplementary Figs. 20–21)[41]. The Zn-HHTP-HS and Zn-HHTP-B films share identical crystalline structures with Zn-HHTP-H (Supplementary Fig. 22). Fourier transform infrared (FT-IR) spectroscopy, X-ray photoelectron spectroscopy (XPS) and thermogravimetric analyses (TGA)

further confirmed the identical compositions of these Zn-HHTP films (Supplementary Figs. 23–25). The Brunauer−Emmett−Teller (BET) measurements revealed that the surface areas of the Zn-HHTP-H (614.0 m² g⁻¹) and Zn-HHTP-HS (479.4 m² g⁻¹) films were much higher than that of the Zn-HHTP-B film (165.3 m² g⁻¹), although they all possessed 1.1 nm micropores (Fig. 3e and Supplementary Fig. 26).

To determine the difference in the gas permeability, the abovementioned Zn-HHTP films were further synthesized on a porous nylon 66 membrane via the MOF-to-*c*-MOF transformation approach (Supplementary Fig. 27). As shown in the permeability test, although the three Zn-HHTP film samples presented a similar film thickness of ~500 nm, much different gas permeabilities were observed (Supplementary Fig. 28). The $N_2$ fluxes of the Zn-HHTP-H film (70.65 L m⁻²s⁻¹) and the Zn-HHTP-HS film (19.21 L m⁻²s⁻¹) were 8.4 and 2.3 times higher than that of the bulk-type Zn-HHTP film (8.45 L m⁻²s⁻¹), respectively (Fig. 3f). Apparently, the hollow interior of the Zn-HHTP films prevented the pressure loss along the path through the *c*-MOF films, leading to high permeability of the *c*-MOF films[42]. Next, the theoretical pressure-driven flow was analyzed to understand the film permeability via commercial CFD software Ansys Fluent (Fig. 3g–i, Supplementary Figs. 29, 30). The simulated $N_2$ fluxes of the Zn-HHTP-H and Zn-HHTP-HS films were 10.2 and 2.2 times higher than that of the Zn-HHTP-B film, respectively, consistent with the experimental permeability results (Fig. 3g–i).

## Numerical simulations of the film structure−gaseous fluid interaction

The flow in the vicinity of the solid surface was facilitated by introducing hollow cavities into the bulk-type Zn-HHTP films (Fig. 4, Supplementary Figs. 31, 32 and Supplementary Table 3). Mass transfer mainly depended on the slow molecular diffusion onto the surface of bulk-type film, while extra nonzero convection velocities were generated on the hollow film (Fig. 4a). The

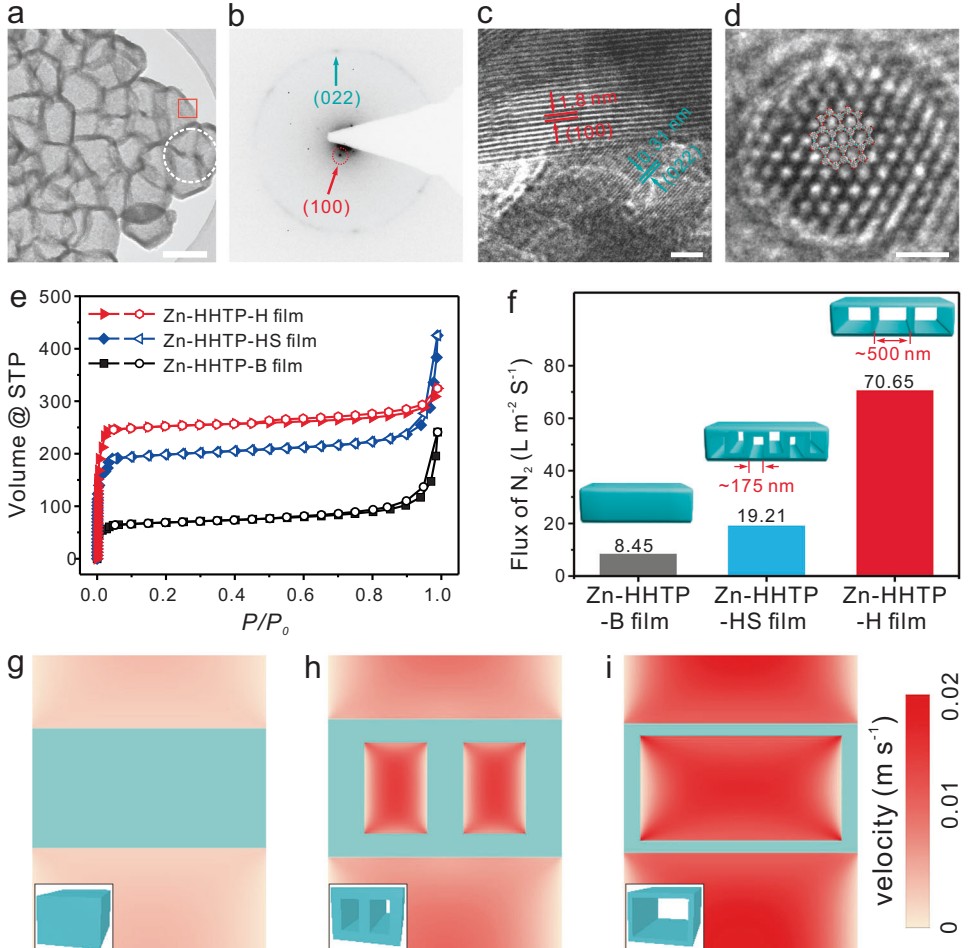

**Fig. 3 | Characterization and gas permeability properties. a** TEM image of the Zn-HHTP-H scraped off from the film. **b** SAED pattern (white circle in (**a**)). **c**, **d** High-resolution TEM image of the red square shown in (**a**). **e** Nitrogen adsorption and desorption isotherms measured at 77 K for different Zn-HHTP films. **f** Comparison of the $N_2$ flux values of different Zn-HHTP films at the same operating pressure. **g**–**i** Contour plots of velocity through the film in the plane of symmetry. The insets show the models for three different Zn-HHTP films. The scale bars represent 500 nm for (**a**) and 10 nm for (**c**, **d**).

surface convection induced by the nonzero velocity enhanced the mass transport, thus largely reducing the time required to achieve equilibrium concentration fields for the hollow films (Fig. 4b, Supplementary Videos 1–3). The simulation results suggested that the mass transfer efficiency followed the order Zn-HHTP-B film < Zn-HHTP-HS film < Zn-HHTP-H film.

The average surface velocity was positively related to the hollow volume $V_{hollow}$ (Fig. 4c). The MOF film models were built with three different hollow volume ratios $\beta = V_{hollow}/V_{bulk}$ ($\beta = 0$ for the Zn-HHTP-B film, $\beta = 0.43$ for the Zn-HHTP-HS film, and $\beta = 0.76$ for the Zn-HHTP-H film; $V$ is the volume of the film). To quantify the influence of convection velocity on the mass transfer efficiency, $U_s$ was obtained from the top surface area ($A$) and the local velocity ($u$) on the surface of MOF films by applying the equation $U_s = 1/A \int u\,dS$ ($dS$ is the unit area in integral). The values of $U_s$ were then calculated to be $4.87 \times 10^{-6}$ m/s, $1.43 \times 10^{-5}$ m/s, and $3.94 \times 10^{-5}$ m/s for the Zn-HHTP-B, Zn-HHTP-HS, and Zn-HHTP-H films, respectively, indicating that the Zn-HHTP-H film allowed higher mass transfer via convection (Fig. 4b). Such high mass transport of the Zn-HHTP-H film was attributed to the high permeability within the hollow film (Fig. 3f). Along with the high permeability, the surface convection was increased. The increased convection on the hollow film surface in turn enlarged

the concentration gradient and enhanced the transport from the environment to the film surface.

Furthermore, we proposed a model to express the relationship between the surface fluid behavior and the hollow nature of the Zn-HHTP porous media (see the "Methods"). By applying the formula derived by Beavers and Joseph[43], we implemented a nonzero velocity over the classical bulk-type porous films as the slip boundary condition to solve the governing equation of Stokes flow (see the "Methods"). The velocity U in the fluid region above the film was described as follows:

$$U = -\frac{1}{2\mu}\left[\frac{H^2 + 2K_{bulk}\alpha H}{1 + \alpha H} - \left(\frac{2K_{bulk}\alpha - \alpha H^2}{1 + \alpha H}\right)z - z^2\right]\frac{dp}{dx} \quad (5)$$

where $K_{bulk}$ is the permeability, $\mu$ is the dynamic viscosity, $\alpha$ is the slip coefficient, $H$ represents the height of the fluid region, $z$ is the distance between the position and the film surface, and $-dp/dx$ is the pressure gradient.

The flow in the Zn-HHTP-H and Zn-HHTP-HS channels was supposed to still follow Eq. (5), and the morphological hollow was regarded as a porosity factor of the porous medium[44,45]. Herein, we applied the generalized Kozeny–Carman (KC) model to derive the hollow-

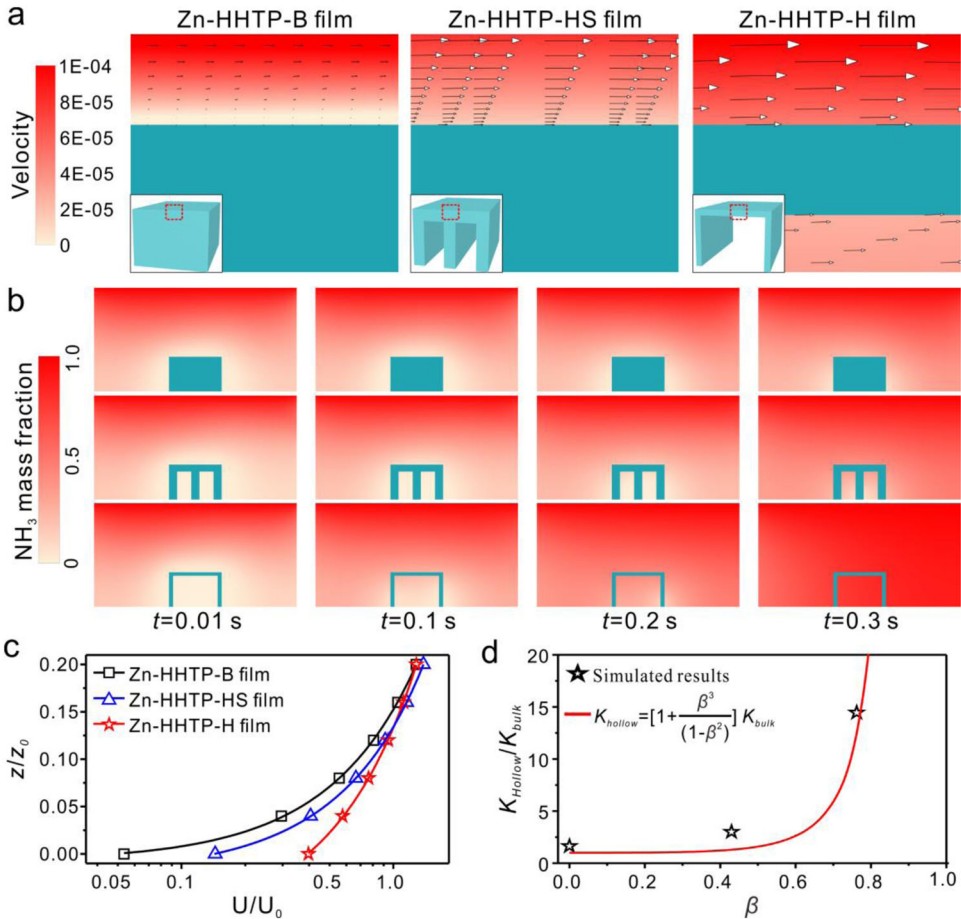

**Fig. 4 | Numerical simulations of the flow velocity and mass transfer. a** Vectors and magnitudes of velocities in the plane of symmetry. The inset images show the models for three different Zn-HHTP films. **b** Time evolutions of NH₃ concentration distributions (from 0 to 0.3 s). **c** The average surface velocities U in different planes above the surface of the Zn-HHTP films. $U_0$ and $z_0$ represent the average inlet velocity and the thickness of the films, respectively. **d** The theoretical permeability $K_{Hollow}$ for the Zn-HHTP films with varying hollow volume ratios $β$.

related permeability $K_{Hollow}$ (see the "Methods")[46,47].

$$U = -\frac{1}{2\mu}\left[\frac{H^2 + 2K_{hollow}\alpha H}{1+\alpha H} - \left(\frac{2K_{hollow}\alpha - \alpha H^2}{1+\alpha H}\right)z - z^2\right]\frac{dp}{dx} \quad (6)$$

$$K_{Hollow} = \left[1 + \frac{\beta^3}{(1-\beta)^2}\right] \quad (7)$$

The contribution of internal structural voids to the enhancement of mass transport near the films at varying hollow volume ratios was determined by Eqs. (6) and (7). By fitting the simulated velocity based on the function form of Eq. (6), the values of $K_{Hollow}$ for the Zn-HHTP-H and Zn-HHTP-HS models were obtained (Fig. 3c and Supplementary Table 4). The corresponding results from the modified permeability model, i.e., Eq. (7), indicated that the present simulation was in good agreement with the gas permeability experimental results for the Zn-HHTP film (Figs. 3f, 4d and Supplementary Table 4). The proposed model was found to be reliable for evaluating the surface fluid behavior of porous films with internal hollow structures, which offered guidance to further control and optimize the overall mass transfer performance.

## Chemiresistive gas-sensing performance

The fresh Zn-HHTP films on silicon substrates were integrated into a chemiresistive device by depositing silver electrodes onto the surface of Zn-HHTP films (Fig. 5a, b and Supplementary Fig. 33). Ammonia (NH₃), a typical biomarker for diseases and a highly toxic and explosive gas in the manufacturing and chemical industries, was used as a molecular probe to elucidate the film structure-dependent mass transport behavior in gas sensing[48,49]. The gas sensing on c-MOF chemiresistive sensors was attributed to the transfer of electrons caused by surface adsorption (Fig. 5a). The n-type dopant of NH₃ can donate electrons to the electron-dominated Zn-HHTP semiconductor[50] (Seebeck coefficient = −4.4 ± 0.07 μV K⁻¹, thermoelectric measurement shown in Supplementary Fig. 34), causing a change in the resistance of the Zn-HHTP sensors (Supplementary Fig. 35). Significantly different response times were observed for different Zn-HHTP film samples, although all the Zn-HHTP samples possessed similar film thicknesses of ~500 nm (Fig. 5c and Supplementary Fig. 36). Upon exposure to 50 ppm ammonia, the Zn-HHTP-H film-based sensor displayed a very fast response of 9.1 s. The corresponding response times of the Zn-HHTP-HS (41.9 s) and Zn-HHTP-B (99.3 s) films were significantly longer than that of the Zn-HHTP-H film. Note that the fast response speed of the Zn-HHTP-H film sensor was superior to that of reported chemiresistive ammonia sensors at room temperature (Fig. 5d and Supplementary Table 5). This excellent response speed indicated that the increased interfacial mass transfer is a dominant factor that greatly promotes the gas-sensing performance.

The Zn-HHTP-H sensor exhibited a response intensity of 43.8% towards 50 ppm ammonia, while the responses of the Zn-HHTP-HS and Zn-HHTP-B films were only 24.8% and 10.8%, respectively

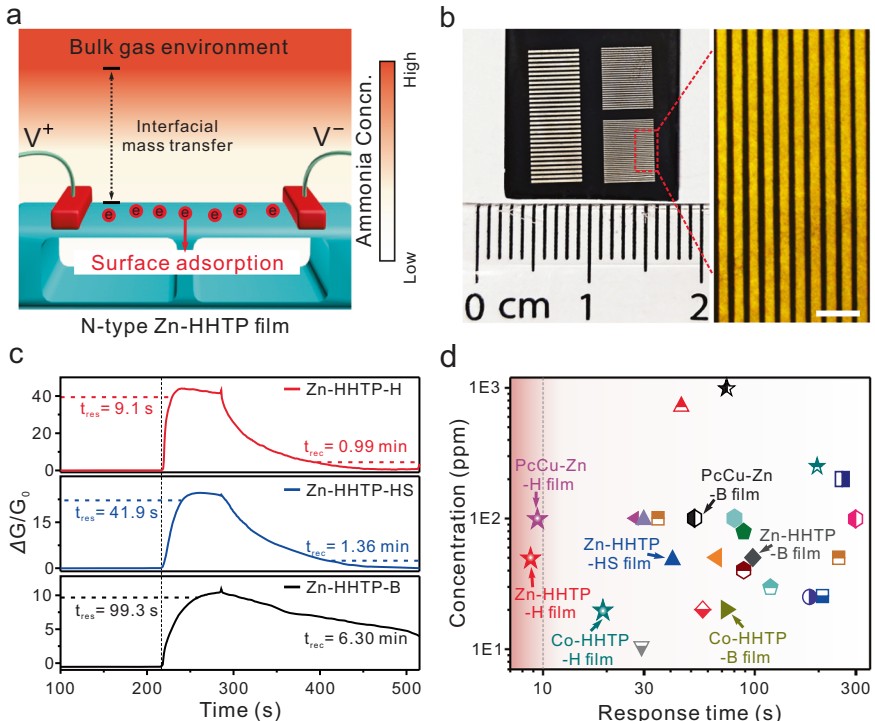

**Fig. 5 | Gas sensing performance. a** Schematic overview of the sensing mechanism of Zn-HHTP film-based sensor towards ammonia. **b** Optical photograph and microscopy image of the fabricated Zn-HHTP-H thin-film-based sensor device. The black area represents the silicon wafer covered by the Zn-HHTP film, and the striped pattern represents the silver electrode. **c** Response and recovery times for three Zn-HHTP films (Zn-HHTP-B, Zn-HHTP-HS, and Zn-HHTP-H films, respectively) based sensors towards 50 ppm ammonia. **d** Comparison of the room-temperature ammonia sensing performance of various chemiresistive materials, such as *c*-MOFs, polymers, and metal oxides (see Supplementary Table 5 for details). The scale bar represents 500 μm for (**b**).

(Supplementary Fig. 36). The theoretical limit of detection (LOD) of the Zn-HHTP-H sensor was calculated to be 39.9 ppb (Supplementary Fig. 37 and Table 5), which was also much lower than those of the other two Zn-HHTP sensors (108.2 ppb for Zn-HHTP-HS and 328.8 ppb for the Zn-HHTP-B film). The anti-interference performance of the Zn-HHTP-H film sensor towards 20 ppm of 12 typical interfering gases is shown in Supplementary Fig. 38a. The results demonstrate that the Zn-HHTP film sensing platform was free from the interference of common gases in practical applications and showed high selectivity towards ammonia. In addition, the Zn-HHTP film exhibited good repeatability with a low coefficient of variation (1.96%) for repeated detection over 20 cycles (Supplementary Fig. 38b).

### Generality of the HSAID strategy

We also synthesized contrast hollow porous *c*-MOF films with different metal sites ($Zn^{2+}$ and $Co^{2+}$) and ligands (PcCu-$(OH)_8$ and HHTP) (PcCu-Zn-H and Co-HHTP-H, Fig. 6a–c). The lattice fringes at interplanar spacings of 1.7 nm and 1.8 nm both corresponded to the (100) planes of PcCu-Zn and Co-HHTP crystals (Supplementary Fig. 39)[23,27]. Although these materials shared the same crystal structures and compositions, the BET measurements indicated that the hierarchical PcCu-Zn-H (677.4 $m^2 g^{-1}$) and Co-HHTP-H (637.9 $m^2 g^{-1}$) films exhibited much higher surface areas than the corresponding bulk-type films (187.0 and 234.2 $m^2 g^{-1}$ for PcCu-Zn-B and Co-HHTP-B, respectively) (Supplementary Figs. 40–47). The gas permeability values of hierarchical *c*-MOF films on nylon 66 membranes (103.94 and 37.35 $L m^{-2} s^{-1}$ for PcCu-Zn-H and Co-HHTP-H, respectively) were obviously superior to those of the contrast bulk-type films (21.81 and 4.85 $L m^{-2} s^{-1}$ for PcCu-Zn-B and Co-HHTP-B, respectively). Figure 6b–d and Supplementary Figs. 48, 49).

Upon exposure to 100 ppm ammonia, the PcCu-Zn-H film-based sensor displayed a very fast response time of 9.8 s, while the

corresponding response time of PcCu-Zn-B (over 53.1 s) was significantly longer (Fig. 6e and Supplementary Fig. 50). Similarly, upon exposure to 20 ppm ammonia, the response speed of the Co-HHTP-H film-based sensor (19.2 s) was much higher than that of the Co-HHTP-B film (over 75.1 s) (Fig. 6f and Supplementary Fig. 51). Such a fast sensing response in hierarchical *c*-MOF film-based sensors was attributed to the hollow nature that promoted interfacial mass transfer between the gas environment and the solid *c*-MOF films.

## Discussion

In summary, taking hierarchical porous *c*-MOF films as models, we demonstrate a general HSAID strategy to efficiently enhance interfacial mass transfer to boost surface reactions (e.g., gas sensing). Hierarchical *c*-MOF films were constructed via a facile transformation from insulating 3D MOFs and possessed well-defined hollow nanostructures. The CFD simulations and permeability test revealed that the hollow nature significantly enhanced the permeability of the hierarchical *c*-MOF film, leading to an increase in the motion velocity of molecules. In particular, the Zn-HHTP-H film displayed a more than 7-fold improvement in local mass transport at the film-gaseous environment interface compared with the Zn-HHTP-B film. As a proof-of-concept model, the Zn-HHTP-H film-based chemiresistive sensor exhibited a 10 times faster response speed toward ammonia than the bulk film, was also superior to the other reported sensors at room temperature. Our work addressed the limit of interfacial mass transport based on an emerging class of conductive and porous MOF films and shed light on the microscopic material–environment interplay and interfacial dynamic mechanisms. Furthermore, this work is also applicable to an enormous range of heterogeneous interactions from simple adsorption to complex catalytic reactions, thus revolutionizing the material design for heterogeneous reactions

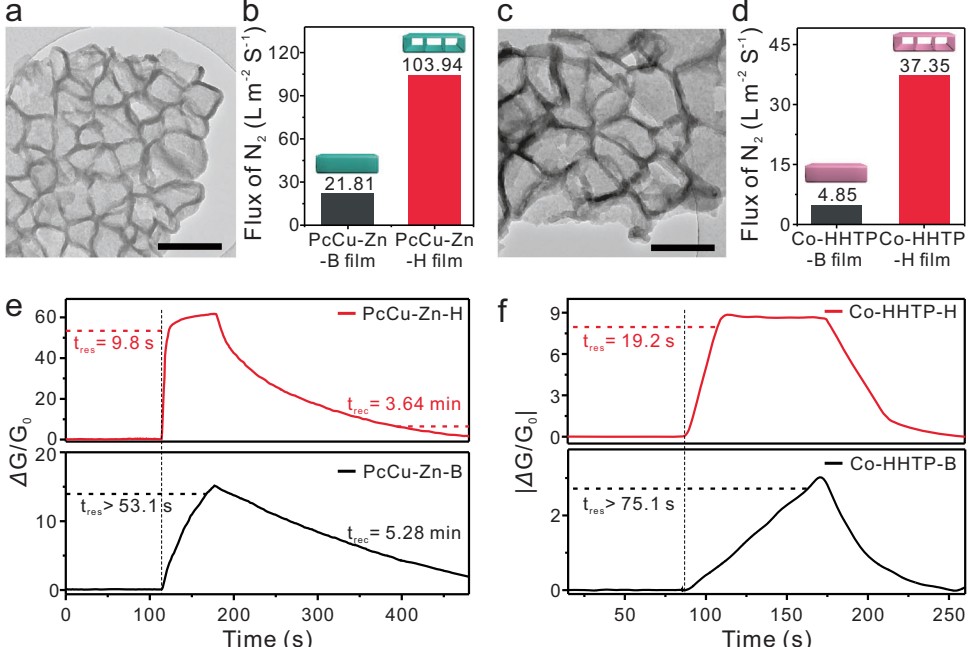

**Fig. 6 | Extended hierarchical MOF films and gas sensing performance. a** TEM image of PcCu-Zn-H scraped off from the film. **b** Comparison of the N$_2$ flux values of the PcCu-Zn-H and PcCu-Zn-B films. **c** TEM image of Co-HHTP-H scraped off from the film. **d** Comparison of the N$_2$ flux values of the Co-HHTP-H and Co-HHTP-B films. **e** Response and recovery times for PcCu-Zn-H and PcCu-Zn-B film-based sensors towards 100 ppm ammonia. **f** Response and recovery times for Co-HHTP-H and Co-HHTP-B film-based sensors towards 20 ppm ammonia. The scale bars represent 500 nm for (**a**) and (**c**).

with highly enhanced efficiency and minimum energy and time consumption.

## Methods

### Preparation of the ZIF-8 film on silicon wafers

A ZIF-8 film with a thickness of ~500 nm was synthesized on a silicon substrate as described previously with some modifications[36]. Typically, the Si/SiO$_2$ wafers were hydroxylated by immersing in a piranha solution (98% H$_2$SO$_4$ and 30% H$_2$O$_2$ (v/v = 3:1)) at 110 °C for 2 h. After rinsing and ultrasonically cleaning three times with pure methanol, a Si/SiO$_2$ wafer (4 × 1.5 cm) was first immersed into 30 mL of 50 mM Zn(NO$_3$)$_2$ • 6H$_2$O methanolic solution under mild stirring for 30 min. Then, 30 mL of 100 mM 2-methylimidazole (Hmim) methanolic solution was added, and the mixture was allowed to react at room temperature for 5 h. The above steps were repeated twice to realize full coverage of the substrate. Finally, the thin films were dried at 80 °C for further use.

### Transformation of ZIF-8 film into Zn-HHTP films

To prepare Zn-HHTP films with different morphologies, a Si/SiO$_2$ wafer (4 × 1.5 cm) with a ZIF-8 film was reacted with 30 mL of an HHTP ethanol/water solution for 24 h. The Zn-HHTP-H films were synthesized by reacting the ZIF-8 film with a 1 mg mL$^{-1}$ solution of HHTP in ethanol/water solution (7:1, v/v) at room temperature. The Zn-HHTP-HS films were synthesized by reacting a ZIF-8 film with a 2 mg mL$^{-1}$ solution of HHTP in ethanol/water solution (5:3, v/v) at 40 °C. The Zn-HHTP NWs were synthesized by reacting a ZIF-8 film with a 2 mg mL$^{-1}$ solution of HHTP in ethanol/water solution (2:6, v/v) at 40 °C. The obtained films were washed with methanol and dried at 80 °C for further use.

### Transformation of a ZIF-8 film into PcCu-Zn-H films

PcCu-(OH)$_8$ was synthesized as described previously[27]. To prepare PcCu-Zn-H films with hollow nanostructures, 28 mL of a 1 mg ml$^{-1}$ PcCu-(OH)$_8$ solution and 120 mg of sodium acetate in DMF/water

solution (7:1, v/v) were prepared first. Next, a Si/SiO$_2$ wafer (4 × 1.5 cm) with a ZIF-8 film was immersed into the solution and reacted at room temperature for 24 h. The obtained PcCu-Zn-H films were washed with methanol and dried at 80 °C for further use.

### Preparation of the ZIF-67 film on silicon wafers

A ZIF-67 film with a thickness of ~500 nm was synthesized on a silicon substrate via a procedure similar to that used for the preparation of the ZIF-8 film. Typically, the hydroxylated Si/SiO$_2$ wafers (4 × 1.5 cm) were first immersed into 15 mL of 25 mM Co(NO$_3$)$_2$ • 6H$_2$O methanolic solution under mild stirring for 10 s. Then, 25 mL of 80 mM 2-Hmim methanolic solution was added, and the mixture was allowed to react at room temperature for 3 h. The above steps were repeated twice to realize full coverage of the substrate. Finally, the thin films were dried at 80 °C for further use.

### Transformation of ZIF-67 film into Co-HHTP-H films

To prepare Co-HHTP films with hollow nanostructures, a Si/SiO$_2$ wafer (4 × 1.5 cm) with ZIF-67 film was reacted with 32 mL of a 1 mg ml$^{-1}$ HHTP ethanol/water (6:2, v/v) solution at room temperature for 24 h. The obtained films were washed with methanol and dried at 80 °C for further use.

### Construction of Zn-HHTP-B films

Zn-HHTP-B films on hydroxylated Si/SiO$_2$ wafers were synthesized as previously reported with minor modification[41]. Typically, zinc acetate (2.9 mmol, 45 mg) was first dissolved in 10 mL of deionized water to prepare solution A in a 20 mL vial. Then, a hydroxylated Si/SiO$_2$ wafer was immersed into solution A under mild stirring for 15 min. The HHTP ligand (1.5 mmol, 48.6 mg) was dispersed in 10 mL of isopropanol to prepare solution B. Solution B was added to solution A, and the mixture was allowed to react at 85 °C for 12 h. The above steps were repeated twice and the obtained films were washed with methanol and dried at 80 °C for further use.

## Construction of PcCu-Zn-B films

PcCu-Zn-B films on hydroxylated Si/SiO$_2$ wafers were synthesized as previously reported with minor modification[41]. Typically, PcCu-(OH)$_8$ (15 mg) was first dissolved in 8 mL of DMF to prepare solution A in a 20 mL vial. Then, ammonia solution (25% v/v, 1 mL) in deionized water (8 mL) and 12 mg Zn(acac)$_2$ were added successively. After sonication for 5 min, a hydroxylated Si/SiO$_2$ wafer was immersed into solution, and the mixture was allowed to react at 65 °C for 12 h. The above steps were repeated twice and the obtained films were washed with methanol and dried at 80 °C for further use.

## Construction of Co-HHTP-B films

Co-HHTP-B films on hydroxylated Si/SiO$_2$ wafers were synthesized with a procedure similar to that used for the preparation of Zn-HHTP films. Typically, cobalt acetate (50 mg) was first dissolved in 10 mL of deionized water to prepare solution A in a 20 mL vial. Then, a hydroxylated Si/SiO$_2$ wafer was immersed into solution A under mild stirring for 15 min. The HHTP ligand (35 mg) was dispersed in 10 mL of isopropanol to prepare solution B. Solution B was added to solution A, and the mixture was allowed to react at 85 °C for 12 h. The above steps were repeated twice and the obtained films were washed with methanol and dried at 80 °C for further use.

## Computational fluid dynamics for gas adsorption in porous media

Three-dimensional numerical simulations of N$_2$/NH$_3$ mixture gas are conducted using ANSYS Fluent 19.0 with a laminar flow model. NH$_3$ gas convection/diffusion and adsorption in Zn-HHTP were modeled with a species transport model combined with a porous media model and user-defined functions (UDFs) to consider the adsorption of NH$_3$ on porous surfaces. Detailed parameters and methods for the simulation are presented in the Supplementary Methods.

We considered the Stokes flow of a viscous fluid through a parallel channel formed by an impermeable upper boundary ($z = H$) and a permeable lower boundary ($z = 0$). The nonslip boundary was applied at the upper boundary, i.e., $u = 0$ at $z = H$, where $H$ represents the height of the fluid region above the film. The film surface velocity U$_s$ was proportional to the shear rate at the permeable boundary, i.e., $\frac{du}{dz} = a(\text{U}_s - \text{U}_D)$ at $z = 0$. Here, $\alpha$ is the slip coefficient, and Darcy's velocity is given by U$_D = -\frac{K}{\mu}\frac{dp}{dx}$, where $dp/dx$ is the pressure gradient.

For the Stokes flow, $\nabla p = \mu \nabla^2 \boldsymbol{u}$, $\frac{dp}{dx} = \mu \frac{\partial^2 u}{\partial z^2}$. According to the boundary conditions given above, the solution in the fluid channel is U $= -\frac{1}{2\mu}\left[\frac{H^2 + 2K_{bulk}\alpha H}{1 + \alpha H} - \left(\frac{2K_{bulk}\alpha - \alpha H^2}{1 + \alpha H}\right)z - z^2\right]\frac{dp}{dx}$.

Thus, the film surface velocity at $z = 0$ can be written as U$_s = -\frac{1}{2\mu}\left(\frac{H^2 + 2K_{bulk}\alpha H}{1 + \alpha H}\right)\frac{dp}{dx}$.

The derivation process of the permeability-porosity relationship is presented below. We considered a film consisting of circular pores with a cross-sectional area of $A_i$ and radius of $r_i$. The total cross-sectional area $A$ of the film was $\int A_i dN(r_i)$, where $dN(r_i)$ denotes the number of pores with a radius of $r_i$.

For laminar flow, U$_i(r_i) = -\frac{r_i^2}{8}\frac{1}{\mu}\frac{dp}{dx}$, where U$_i(r_i)$ is the average velocity in individual pores. The average velocity through the cross-sectional area $A$ was approximated as U$_f = \frac{1}{A}\int \text{U}_i(r_i)A_i dN(r_i) \approx \frac{\sum \text{U}_i(r_i)A_i dN(r_i)}{\sum A_i dN(r_i)} \propto \varepsilon \text{U}_i(r_h)$, where the hydraulic radius $r_h \propto \frac{\varepsilon}{(1-\varepsilon)}$ was introduced to replace $r_i$. Thus, U$_f \propto -\varepsilon\frac{r_h^2}{8}\frac{1}{\mu}\frac{dp}{dx} = -\frac{\varepsilon^3}{(1-\varepsilon)^2}\frac{1}{\mu}\frac{dp}{dx}$. A comparison of U$_f$ with Darcy's velocity yielded the permeability-porosity relationship $K_{bulk} = C_{kc}\frac{\varepsilon^3}{(1-\varepsilon)^2}$.

We believe the permeability-porosity relationship derived above is still valid for hollow interior films. It is reasonable to assume that the contribution of hollows by volume ratio is directly analogous to the effect of porosity on permeability, and the difference in permeability caused by hollows is proportional to the term $\frac{\varepsilon^3}{(1-\varepsilon)^2}$. Therefore, we substituted the hollow volume ratio $\beta$ for the porosity $\varepsilon$ and directly added an increment $\triangle K_{Hollow} = \frac{\beta^3}{(1-\beta)^2}K_{bulk}$ to the original $K_{bulk}$. Finally, we obtained the permeability of the hollow interior film from the following equation: $K_{Hollow} = \left[1 + \frac{\beta^3}{(1-\beta)^2}\right]K_{bulk}$.

The values of $K_{Hollow}$ for the Zn-HHTP-H and Zn-HHTP-HS models were calculated using the data in Fig. 4c (for details, see Supplementary Table 3). Here, $H$ is 1 μm, and $K_{bulk}$ was set to $2 \times 10^{-15}$ m$^2$. For example, according to Eq. (5) for the Zn-HHTP-HS film, $\frac{H^2 + 2K_{Hollow}\alpha H}{1 + \alpha H} = 1.36 \times 10^{-14}$ m$^2$, $\frac{2K_{Hollow}\alpha - \alpha H^2}{1 + \alpha H} = -1.28 \times 10^{-6}$ m$^2$, $K_{Hollow}$ was calculated to be $6.01 \times 10^{-15}$ m$^2$. For the Zn-HHTP-H film, $\frac{H^2 + 2K_{Hollow}\alpha H}{1 + \alpha H} = 7.06 \times 10^{-14}$ m$^2$, $\frac{2K_{Hollow}\alpha - \alpha H^2}{1 + \alpha H} = -1.69 \times 10^{-6}$ m$^2$, and $K_{Hollow}$ was calculated to be $2.88 \times 10^{-14}$ m$^2$. $K_{Hollow}/K_{bulk}$ were thus calculated to be 3.01 and 14.45 for the Zn-HHTP-HS and Zn-HHTP-H films in our simulations, respectively. Additionally, according to Eq. (7) with $\beta = 0.43$ and 0.76, $K_{Hollow}/K_{bulk}$ was calculated to be 1.25 and 8.81 for the Zn-HHTP-H and Zn-HHTP-HS films, respectively.

## Permeability test

The permeance test was conducted by means of a pressure filtration device (HP4750, Sterlitech). The 2D c-MOF films coated on a nylon 66 membrane with a diameter of 22 mm were used for testing. The operating temperature was 25 °C, and the operating pressure was 0.5 bar. A soap bubble flowmeter was used to measure the gas flux.

## Gas sensing experiments

The ~500 nm thick 2D c-MOF film samples on Si/SiO$_2$ wafers were integrated into a chemiresistive device by depositing silver electrodes with a channel length of 80 μm. The fabricated sensors were aged at room temperature in air for 6 h. The fabricated sensors were aged at room temperature in air for 6 h. The sensing tests were carried out on a CGS-4TPs (Beijing Elite Tech Co., Ltd., China) intelligent sensing analysis system. The relative humidity (RH) was 25 ± 10%. Ammonia gases with different concentrations are generated by dropping certain amounts of ammonia water (25–28 wt%) with a microsyringe onto an evaporator in the test chamber (total volume of 20 L). Other kinds of gases, including ethanol, acetone, toluene, formaldehyde, and water, that are similar to ammonia were used. Detailed methods and experimental setup for the gas concentration regulation and gas-sensing test are presented in the Supplementary Methods.

## Reporting summary

Further information on research design is available in the Nature Portfolio Reporting Summary linked to this article.

## Data availability

Raw data for all figures, plots, and particle size distributions are available from the corresponding author upon request.

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

## Acknowledgements

This work was financially supported by EU Graphene Flagship (GrapheneCore3 881603), CRC 1415 (Chemistry of Synthetic Two-Dimensional Materials, No. 417590517), as well as the German Science Council and Center of Advancing Electronics Dresden (cfaed). This project also has received funding from the European Research Council (ERC) under the European Union's Horizon 2020 research and innovation program (FC2DMOF, grant agreement No. 852909). R.D. thanks Taishan Scholars Program of Shandong Province (tsqn201909047) and National Natural Science Foundation of China (22272092). C.H. gratefully acknowledges funding from the Alexander von Humboldt Foundation. The authors gratefully acknowledge Dr. Li Ding at School of Chemistry and Chemical Engineering, South China University of Technology for the gas permeability test. The authors thank Dr. Xiangyu Chen at Institute of Chemistry, Chinese Academy of Sciences for the XPS measurement, as well as Dr. Zichao Li for the conductivity testing. We acknowledge Dresden Center for Nanoanalysis (DCN) at TUD and Dr. Petr Formanek (Leibniz Institute for Polymer Research, IPF, Dresden) for the use of facilities.

## Author contributions

R.D. and X.F. proposed the research direction and guided the project. C.H. designed and performed the experiments (materials synthesis, structure and morphology characterization). X.S. and G.H. designed and performed the theoretical calculation. X.Z., Z.Z., T.W., and S.M. designed and performed gas sensing testing. M.L., Z.L., H.Q., U.K., D.S., and A.F. contributed to the characterizations. X.H., Y.L., and M.W. joined the discussion of data and gave useful suggestions. R.D., C.H., and X.F. analyzed the experimental results and drafted the manuscript with the input from all the co-authors.

## Funding

## Competing interests

The authors declare no competing interests.

## Additional information

[1]Center for Advancing Electronics Dresden (Cfaed) and Faculty of Chemistry and Food Chemistry, Technische Universität Dresden, 01062 Dresden, Germany. [2]Department of Engineering Mechanics & State Key Laboratory of Fluid Power and Mechatronic Systems, Zhejiang University, Hangzhou 310027, China. [3]Tianjin Key Laboratory of Drug Targeting and Bioimaging, Life and Health Intelligent Research Institute, Tianjin University of Technology, Tianjin 300384, People's Republic of China. [4]Center for Advancing Electronics Dresden (cfaed) and Faculty of Electrical and Computer Engineering, Technische Universität Dresden, 01062 Dresden, Germany. [5]Dresden Center for Nanoanalysis, Center for Advancing Electronics Dresden, Technische Universität Dresden, 01062 Dresden, Germany. [6]Fraunhofer Institute for Ceramic Technologies and Systems (IKTS), Maria-Reiche-Strasse 2, 01109 Dresden, Germany. [7]Electron Microscopy of Materials Science, Central Facility for Electron Microscopy Universität Ulm, 89081 Ulm, Germany. [8]Leibniz-Institut für Polymerforschung Dresden e.V. (IPF), Hohe Str. 6, Dresden 01069, Germany. [9]Department of Synthetic Materials and Functional Devices, Max Planck Institute for Microstructure Physics, D-06120 Halle (Saale), Germany. [10]Key Laboratory of Colloid and Interface Chemistry of the Ministry of Education, School of Chemistry and Chemical Engineering, Shandong University, Jinan 250100, China. [11]These authors contributed equally: Chuanhui Huang, Xinglong Shang, Xinyuan Zhou. ✉ e-mail: ghu@zju.edu.cn; xinliang.feng@tu-dresden.de; renhaodong@sdu.edu.cn

