## [Peer Review File · Nature Communications]

Hierarchical conductive metal-organic framework films enabling efficient interfacial mass transferREVIEWER COMMENTS

Reviewer #1 (Remarks to the Author):

In this manuscript, the authors report the fabrication of hollow structured c-MOFs for enhanced interfacial mass transfer and their applications as gas sensors with fast responses. Their synthetic strategy for fabricating hollow c-MOFs is very interesting and useful. However, further explanation is required to support the authors' claims on fast mass transfer in hollow c-MOFs. Also, the delivery of the manuscript is insufficient due to several omissions and errors in the description. In this regard, the following revisions are required for this work to meet the standard of publication on Nat. Commun.

1. The authors explain that interfacial mass transport is faster in hollow structures, assuming that parallel flow is supplied along the surface. However, in gas sensors, analytes can be transported to the surface from all directions, including vertical direction. Is the authors' claim valid under these circumstances? How significant is parallel mass transport along the surface for the speed of real gas-solid reactions?
2. The authors discussed the relationship between enhanced mass transport and porous structure, focusing on the macroscopic hollow volume (β). What about the effect of the size or volume of internal pores in MOFs? As the crystalline porous structure is a unique feature of MOFs, consideration of contribution from micropores of MOFs, together with macroscopic hollow structure, would provide a deeper understanding of the MOF-based porous medium.
3. More details on the sensing measurements setup should be provided. How was the gas concentration regulated from the dropped ammonia water? How was the gas supplied to the sensor, and at what flow rate?
4. Please check the typo and errors in the calculated ΔG value in table S1.
5. While I agree with the faster response speed of hollow samples, the authors should mention that the calculated response time values can be inaccurate in several samples. As some samples did not reach saturation during exposure, the calculated response time will differ depending on gas exposure duration.
6. The authors should provide explanation of Fig. 3g-i in the main text.

Reviewer #2 (Remarks to the Author):

In this manuscript, the authors reported a hierarchical-structure-accelerated interfacial dynamic (HSAID) strategy to efficiently improve interfacial gas transfer on conductive metal-organic framework (c-MOF) films via the construction of hierarchical porous structures. In addition, the Zn-HHTP-H film was integrated into a chemiresistive ammonia sensor and showed good performance. These innovative results are logically presented and would raise wide interest. This work is suitable for publication in Nature communication after addressing the following issues.

1. The authors only supplied the photo of ZIF-67 film coated on Si/SiO wafer (Fig. S5). To make direct comparisons, the photographs of Zn-HHTP-HS and Zn-HHTP-B films on Si/SiO₂ wafer should be supplied in the manuscript.
2. There are still some spelling and grammar mistakes in this manuscript. Please read carefully and correct all of them, such as on page S5, the unit of the aging time was missing, which would confuse other researchers repeating this experiment.
3. On page 6, Fig. 2c-g don't match the description 'All these hierarchical c-MOF films exhibited a highly crystalline structure and intrinsic electrical conductivity, while the inside was hollow', please check carefully and make a correction.
4. On page 7, Supplementary Fig. S9b, 9d, 9e cannot match the contents in support information, please check carefully and make a correction.
5. On page S20, the authors claim that HHTP could not be protonated, when only organic solvents

existed such as EtOH or DMF. This statement should be made a reconsideration. The method of sacrificial ZIF-template for MOF film conversion is the key part of this article, please discuss in detail about the water and solvent factors of this process.

6. Several important literatures about MOFs in heterogeneous catalysis and sensing should be cited in the manuscript, such as *Angew.Chem.Int.Ed.* 2021, 60, 10806–10813; *J. Am. Chem. Soc.* 2021, 143, 12129–12137.

RESPONSE TO REVIEWERS' COMMENTS

Response To Reviewer 1:

General comment: In this manuscript, the authors report the fabrication of hollow structured c-MOFs for enhanced interfacial mass transfer and their applications as gas sensors with fast responses. Their synthetic strategy for fabricating hollow c-MOFs is very interesting and useful. However, further explanation is required to support the authors' claims on fast mass transfer in hollow c-MOFs. Also, the delivery of the manuscript is insufficient due to several omissions and errors in the description. In this regard, the following revisions are required for this work to meet the standard of publication on *Nat. Commun.*

Reply: We sincerely appreciate this reviewer for the encouraging comments and the positive recommendation for publication after revision. According to your valuable suggestions, we here further addressed all the issues carefully that make our work of great interest to the broad readership of *Nat. Commun.*

Comment 1: The authors explain that interfacial mass transport is faster in hollow structures, assuming that parallel flow is supplied along the surface. However, in gas sensors, analytes can be transported to the surface from all directions, including vertical direction. Is the authors' claim valid under these circumstances? How significant is parallel mass transport along the surface for the speed of real gas-solid reactions?

Reply: We fully understand the reviewer's concerns about mass transport in the vertical direction. As shown in our schematic of interfacial mass transport on a solid porous film (Fig. 1a and 1b), The concentration gradient-induced molecular diffusion (vertical direction) and the mass transport induced by surface convection (horizontal direction) have both already been taken into account in our study. Herein, based on the systematic numerical simulations and experimental results, it can be concluded that the interfacial mass transfer mainly depends on the slow molecular diffusion for the bulk-type **Zn-HHTP-B** film. However, for the hierarchical **Zn-HHTP-H** film, the interfacial mass transfer depends on both the molecular diffusion and surface convection induced mass transport. Compared with the **Zn-HHTP-B** film, the enhanced interfacial mass transport on hierarchical hollow **Zn-HHTP-H** films should be attributed to hierarchical

hollow structure induced surface convection (i.e., parallel mass transport along the surface).

Fig. 1. Schematic of interfacial mass transport on a solid porous film.

First, For the interfacial mass transport, two factors were involved (one is the velocity vector of the gas fluid (left in Fig.1), the other is the concentration field of gas molecules (right in Fig.1)). For the concentration field of gas molecules, the gas molecules are indeed transported from the bulk gas environment to the film surface from vertical direction, as mentioned by the reviewer. However, for the velocity vector of the gas fluid, it consists of the concentration gradient-induced molecular diffusion (vertical direction) and the surface convection (V in horizontal direction). However, previous studies mainly focused on the concentration gradient-induced molecular diffusion in the vertical direction (Noboru Yamazoe, et al. *Sens. Actuators. B Chem.* 2001, **80**, 125-131; Noboru Yamazoe, et al. *Sens. Actuators. B Chem.* 2003, **96**, 226-233), which has led to parallel mass transport induced by surface convection being overlooked. Herein, based on our study, the mass transport induced by surface convection was found to be able to greatly enhance the interfacial mass transport.

Second, for the surface fluid behaviour at the boundary layer between the bulk gas environment and porous c-MOFs medium, the surface velocity U was described by the famous Beavers and Joseph theory (*J. Fluid Mech.*, 1967, 30 197–207) as follows:

$$U_s = -\frac{1}{\mu} \left[\frac{H^2 + 2K_{bulk}\alpha H}{2 + 2\alpha H} \right] \frac{dp}{dx}$$

As shown in the computational fluid dynamics simulation, the parallel surface velocity U_s at the film surface is $3.94 \times 10^{-5} \text{ m s}^{-1}$ for the **Zn-HHTP-H** film, which is over one order of magnitude higher than vertical velocities ($\sim 10^{-6} \text{ m s}^{-1}$), indicating the **Zn-**

HHTTP-H film allows higher mass transfer via parallel convection rather than diffusion. Moreover, the Péclet number characterizing the ratio of the parallel advection to the diffusion (Trautz, M., et al. *Ann. Phys.* 1935, **414**, 333-352), i.e., UL/D , is at least one order of magnitude higher for the **Zn-HHTTP-H** film (0.2) than for the **Zn-HHTTP-B** film (0.027). This means that the diffusion dominates the mass transport for the **Zn-HHTTP-B** films. Therefore, we can focus on the enhanced mass transport for the **Zn-HHTTP-H** film by analyzing the parallel convection velocity.

Third, all the Zn-HHTTP films have the same crystalline structures and chemical composition, as demonstrated by the XRD patterns, the FT-IR spectra, XPS, and TGA curves (Fig. S22-S25). The three Zn-HHTTP films possess different sizes of hollow cavities (Fig. S13, S15, S16 and S20), which lead to much different gas permeability (Fig. 3f-3i). Although all the Zn-HHTTP samples possessed similar film thicknesses of ~ 500 nm, the hollow Zn-HHTTP-H (9.1 s) film-based sensor displayed a much faster response speed than those of Zn-HHTTP-HS (41.9 s) and Zn-HHTTP-B (99.3 s). Obviously, for the Zn-HHTTP-B film, the slow molecular diffusion on the surface results in slow mass transfer (slow response speed). After introducing hollow cavities inside the film (Zn-HHTTP-H film), the gas permeability of the film was greatly enhanced and extra nonzero convection velocities were generated on the surface. The increased convection on the hollow film surface in turn enlarged the concentration gradient and enhanced the transport from the environment to the Zn-HHTTP-H film surface.

In conclusion, the faster sensing response in hierarchical c-MOF film than that in the bulk-type film should be attributed to the hollow structure-induced high permeability greatly promotes interfacial mass transfer. We believe that the present results can establish the relationship between hollow induced permeability in porous c-MOF and its interfacial mass transfer performance.

Comment 2: The authors discussed the relationship between enhanced mass transport and porous structure, focusing on the macroscopic hollow volume (β). What about the effect of the size or volume of internal pores in MOFs? As the crystalline porous structure is a unique feature of MOFs, consideration of contribution from micropores of MOFs, together with macroscopic hollow structure, would provide a deeper understanding of the MOF-based porous medium.

Reply: We appreciate the reviewer for keenly pinpointing the major contribution of this work and appreciate this constructive comment. In fact, the effect of the intrinsic micropores on MOFs has already been considered. This research is aimed at

establishing a relationship between the hierarchical structure in porous c-MOFs film and interfacial mass transfer. For such a purpose, the porous nature of the c-MOFs nanofilm itself is the premise, otherwise the macroscopic hollow structure inside the film cannot interact with the external environment.

First, from an experimental point of view, all the Zn-HHTP films have the same crystalline structures and chemical composition, as demonstrated by the XRD patterns (Fig. S22), the FT-IR spectra (Fig. S23), XPS (Fig. S24), and the TGA curves (Fig. S25). Meanwhile, The Brunauer–Emmett–Teller (BET) measurements further revealed that all the Zn-HHTP films possessed the same 1.1 nm micropores (Fig. S26). The only significant difference in the Zn-HHTP films is their macroscopic hollow structure (Fig. S15, S16 and S20), which results in various gas permeability of films and significantly different mass transfer speeds in gas sensing (Fig.3f-3i and Fig.5c). All the results demonstrated that the enhanced mass transport in hierarchical c-MOF films mainly due to the effect of macroscopic hollow structure in the films. The relevant statements were shown on page 7-8 in the revised manuscript and the relevant characterizations are shown on page S31-S35 in the revised Supporting Information.

Second, from an numerical simulations of the film structure–gaseous fluid interaction point of view, in our model, the internal pores of the c-MOFs framework have already been taken into account and were modelled as the porous medium model with a fixed permeability ($2 \times 10^{-15} \text{ m}^2$) and a porosity (0.2 – 0.3, see Supplementary Table S3), while the flow in the hollow is solved directly. For example, the bulk-type Zn-HHTP-B film was modelled as a porous medium with a porosity ($\varepsilon^a = 0.2$) and a BET ($165.3 \text{ m}^2 \text{ g}^{-1}$), which strictly follows the data obtained from the BET gas adsorption measurement. Moreover, K_{hollow} from our modified permeability model consists of the contribution of intrinsic micropores, i.e., K_{bulk} , and hollow structure, i.e., $\frac{\beta^3}{(1-\beta)^2} K_{bulk}$. The permeability of the Zn-HHTP-H film increases by 378% as β increases from 0.43 to 0.76.

The relevant statements and tables are shown on page 17 in the revised manuscript and page S63 in revised Supporting Information.

Comment 3: More details on the sensing measurements setup should be provided. How was the gas concentration regulated from the dropped ammonia water? How was the gas supplied to the sensor, and at what flow rate?

Reply: Thanks for the constructive comments, and we added detailed methods for gas sensing experiments to the revised manuscript and supporting information.

First, the volume of ammonia water for gas concentration regulation was described as follows:

$$V_x = \frac{VCM}{22.4 \times DP} \times 10^{-9} \times \frac{273 + T_R}{273 + T_B}$$

Where, V is the volume of the gas sensing test chamber, which is 20 L; C is the gas concentration, ppm; M is liquid molecular weight; D is the liquid density, g cm^{-3} ; P is the liquid purity; T_R is the room temperature, $^{\circ}\text{C}$; T_B is the temperature inside the gas sensor test chamber, $^{\circ}\text{C}$; V_x is the volume of fluid to be injected.

Taking 50 ppm ammonia gas as an example, V_x is calculated to be 3.2 μL according to the above formula.

Second, the detailed method for the gas supplied to the sensor is following:

Using a micro-syringe to take a certain volume of ammonia water and put it into a high-temperature evaporating dish ($150\text{ }^{\circ}\text{C}$) inside the gas-sensing test chamber. Then ammonia water will evaporate into ammonia gas. Under the continuous agitation of the fan, the ammonia gas quickly would fill the entire test chamber and diffuse to the surface of the sensor. The overall gas flow rate defaults to 20 L min^{-1} .

Third, the diagram of the experimental setup is shown as follows:

Supplementary Fig. S33. (a) Diagram of the gas-sensing test device and (b) schematic diagram of airflow diffusion.

The relevant statements were shown in page 18 in revised manuscript and page S7 and S42 in revised Supporting Information.

Comment 4: Please check the typo and errors in the calculated ΔG value in table S1

Reply: Thanks for the timely comments, and we modified the calculation formula in table S1. The relevant statements were shown in page S61 in Supporting Information.

Comment 5: While I agree with the faster response speed of hollow samples, the authors should mention that the calculated response time values can be inaccurate in several samples. As some samples did not reach saturation during exposure, the calculated response time will differ depending on gas exposure duration.

Reply: Thank you for the reviewer's support of the main point and constructive comments.

For the Zn-HHTTP sensors in Fig. 5c, the response of all the three sensors towards ammonia gas has reached saturation, and the calculation of the response time is based on the time to reach 90% of the response value. But, as the reviewers pointed out, some sensors indeed did not reach saturation in response to ammonia (Fig. 6e and 6f). In Fig. 6e, the response of the hollow Pc-CuZn-H to ammonia has reached saturation, and the calculated response time was 9.8 s, while the response time of bulk Pc-CuZn-B was longer than 53.1s. The situation in Fig. 6f is similar. The response of the hollow Co-HHTTP-H to ammonia has reached saturation, and the calculated response time was 19.2 s, while the response time of the bulk Co-HHTTP-B was longer than 75.1s. All the above results demonstrate that the hierarchical hollow nanostructures would promote interfacial mass transfer, leading to a fast response speed in gas sensing.

Following the suggestion from the reviewer, we have modified Fig. 6e and 6f and the relevant statements in the revised manuscript as following:

Upon exposure to 100 ppm ammonia, the **PcCu-Zn-H** film-based sensor displayed a very fast response time of 9.8 s, while the corresponding response time of **PcCu-Zn-B** (over 53.1 s) was significantly longer (Fig. 6e and Supplementary Fig. S50). Similarly, upon exposure to 20 ppm ammonia, the response speed of the **Co-HHTTP-H** film-based sensor (19.2 s) was much higher than that of the **Co-HHTTP-B** film (over 75.1 s) (Fig. 6f and Supplementary Fig. S51).

The relevant statements were shown on page 14 of the revised manuscript.

Comment 6: The authors should provide explanation of Fig. 3g-3i in the main text.

Reply: Following the suggestion from the reviewer, and the detailed explanation of Fig. 3g-i has been added in the revised manuscript.

Next, the theoretical pressure-driven flow was analysed to understand the film permeability via commercial CFD software FLUENT (Fig. 3g-3i, Supplementary Figs. S29-S30). The simulated N₂ fluxes of the **Zn-HHTP-H** and **Zn-HHTP-HS** films were 10.2 and 2.2 times higher than that of the **Zn-HHTP-B** film, respectively, consistent with the experimental permeability results (Fig. 3g-3i).

The relevant statements were shown on page 8 of the revised manuscript.

Response To Reviewer 2:

General comment: In this manuscript, the authors reported a hierarchical-structure-accelerated interfacial dynamic (HSAID) strategy to efficiently improve interfacial gas transfer on conductive metal-organic framework (c-MOF) films via the construction of hierarchical porous structures. In addition, the Zn-HHTP-H film was integrated into a chemiresistive ammonia sensor and showed good performance. These innovative results are logically presented and would raise wide interest. This work is suitable for publication in Nature communication after addressing the following issues.

Reply: We sincerely appreciate this reviewer for his/her positive comments, recommendations, and detailed suggestions regarding the improvement of our manuscript. Moreover, we here further addressed the issues carefully that make our work of great interest to the broad readership of *Nat. Commun.*

Comment 1: The authors only supplied the photo of ZIF-67 film coated on Si/SiO wafer (Fig. S5). To make direct comparisons, the photographs of Zn-HHTP-HS and Zn-HHTP-B films on Si/SiO₂ wafer should be supplied in the manuscript.

Reply: Following the suggestion from the reviewer, we added the photographs of as-synthesized Zn-HHTP-HS and Zn-HHTP-B films on Si/SiO₂ wafer in Figure S2. Moreover, we added the photographs of as-synthesized PcCu-Zn-B and Co-HHTP-B films on Si/SiO₂ wafer in Figure S5.

The relevant figures are shown in page S10 and S13 in Supporting Information.

Comment 2: There are still some spelling and grammar mistakes in this manuscript. Please read carefully and correct all of them, such as on page S5, the unit of the aging time was missing, which would confuse other researchers repeating this experiment.

Reply: Agreed, we have checked the manuscript and Supporting Information carefully and modified some mistakes.

For the Supporting Information, we have added the unit of the time on page S5, and modified the captions and many relevant statements.

For the manuscript, it has been polished by the native English speaker from Nature Publishing Group Language Editing.

The relevant modifications are shown in the revised manuscript and Supporting Information.

Comment 3: On page 6, Fig. 2c-g don't match the description 'All these hierarchical c-MOF films exhibited a highly crystalline structure and intrinsic electrical conductivity, while the inside was hollow', please check carefully and make a correction.

Reply: We are sorry that we did not provide a correct description of Fig. 2c-g in our previous version. Herein we modify the relevant statements:

Following the same sacrificial template synthetic method, the PcCu-Zn and Co-HHTP films with hierarchical nanostructures were synthesized (named **PcCu-Zn-H** and **Co-HHTP-H**, respectively, Fig. 2c-2g, Supplementary Fig. S5-S8). All these hierarchical c-MOF films exhibited intrinsic electrical conductivity, while the inside was hollow (Fig. 2f-2g, Supplementary Table S2).

The relevant statements are shown on pages 5-6 in the revised manuscript.

Comment 4: On page 7, Supplementary Fig. S9b, 9d, 9e cannot match the contents in support information, please check carefully and make a correction.

Reply: Agreed, and we modified the figure numbers.

The **Zn-HHTP-H** film with ~500 nm cavities was obtained only at low water fractions ($\leq 25\%$) and low temperatures ($\leq 40\text{ }^\circ\text{C}$) (Supplementary Figs. S13b, S13d and S15). When the water fraction and temperature were increased (e.g. 37.5% and 40 $^\circ\text{C}$, respectively), a Zn-HHTP film was formed with small cavities of ~175 nm (denoted as **Zn-HHTP-HS** film, Supplementary Fig. S13b, S13e and S16-S17).

The relevant statements are shown on page 7 in the revised manuscript.

Comment 5: On page S20, the authors claim that HHTP could not be protonated, when only organic solvents existed such as EtOH or DMF. This statement should be made a reconsideration. The method of sacrificial ZIF-template for MOF film conversion is the key part of this article, please discuss in detail about the water and solvent factors of this process.

Reply: we feel sorry about the wrong description on page S20 in our previous version. herein we modify the relevant statements:

First, when only organic solvent was used, no Zn-HHTP product was observed. It was supposed that the generated protons in organic solvent were too less to break the coordination bonds between Zn^{2+} and MeIM linkers in the ZIF-8 crystals. Consequently, the etching step in **equation 2** ($3\text{Zn}(\text{MeIM})_2 + 6\text{H}^+ \rightleftharpoons 6\text{HMeIM} + 3\text{Zn}^{2+}$) was inhibited and finally no Zn-HHTP product was produced. The relevant statements are shown on page S20 in the revised Supporting Information.

Second, according to the constructive comments from the reviewer, herein we added the fundamental understanding of water fraction factors during the transformation process. The conversion process of ZIF-8 to Zn-HHTP depends on two factors, the release rate of Zn^{2+} ions and the coordination rate with HHTP, both of which are greatly affected by the solvent composition and reaction temperature. In our study, systematic experiments were carried out to illuminate the effects of the solvent factors and reaction temperature (Fig. S13a).

Supplementary Fig. S13. (b) Phase diagram that correlates the solvent composition (horizontal ordinate) and reaction temperature (vertical ordinate). (c, d, e, f) Cross-sectional SEM images of the ZIF-8, Zn-HHTP-H, Zn-HHTP-HS and Zn-HHTP nanowire (NW) films, respectively. (g, h) The concentration of Zn^{2+} (g) and pH value (h) versus reaction time toward the construction of Zn-HHTP-H, Zn-HHTP-HS and Zn-HHTP NW films, respectively. Scale bars represent 500 nm for (c-f).

During the whole transformation, the product morphology significantly depends on the reaction kinetics (Supplementary Fig. S13b). At low water fractions and low temperatures, the etching rate of ZIF-8 is supposed to be slowed down and much less than the coordination rate. The diffusion distance of free Zn^{2+} ions is insufficient and

nucleation and growth of Zn-HHTP only happen on the inner wall of the Zn-HHTP shell. When higher water fractions and temperatures are used, the corresponding etching rate increases and is comparable to the coordination rate. In this case, most of the Zn^{2+} ions are preferable to form Zn-HHTP inside the shell, while small part of the Zn^{2+} ions diffuse into the solution and produce Zn-HHTP nanowires outside the shell. If the water fractions and temperatures were further increased, the ZIF-8 etch rate would dramatically increase and faster than the Zn-HHTP shell can form. Therefore, Zn^{2+} ions from the etching of the ZIF-8 nanocrystals mostly diffuse into the solution and react with $(HHTP^*)^{3-}$ ions to produce Zn-HHTP nanowires on the surface of ZIF-8 nanocrystals. The above results underscore that the balance between the etching and coordination rate is pivotal to achieving a well-defined hollow nanostructure.

To verify the above conversion mechanism, the Zn^{2+} concentration and pH value versus reaction time toward the construction of different Zn-HHTP products were monitored (Figures S13g-S13h). The initial Zn^{2+} concentrations in the solution are all around 0.15 ppm. In the case of Zn-HHTP NW, this concentration quickly increased as the reaction was triggered, revealing the rapid generation of free Zn^{2+} ions in the reactions (equation 2). Over the whole transformation, the Zn^{2+} concentration keeps increasing and reaches about 70.0 ppm in equilibrium. For the case of Zn-HHTP-H, the Zn^{2+} concentration presents a very slow growth and is only about 7.7 ppm at equilibrium, which is much lower than those for Zn-HHTP-HS and Zn-HHTP NW. These results indicate that the whole conversion for Zn-HHTP-H takes place inside the Zn-HHTP shell. However, the nucleation and growth of Zn-HHTP NW takes place outside the surface of ZIF-8 with mostly metal ions were released into the solution. To test the change of H^+ concentrations, the pH values were investigated (Supplementary Fig. S13h). The initial pH of HHTP solution is only about 5.3 and the pH values were found to gradually increase over the whole transformation. These results are considered as the protons generated in equation (1) are quickly consumed in equation (2), a trend that is highly consistent with the change of Zn^{2+} concentration.

We have added the above fundamental understanding shown on page S21-S22 in the revised Supporting Information and hope our additional efforts appropriately address your concerns.

Comment 6: Several important literatures about MOFs in heterogeneous catalysis and sensing should be cited in the manuscript, such as *Angew. Chem. Int. Ed.* 2021, 60, 10806–10813; *J. Am. Chem. Soc.* 2021, 143, 12129–12137.

Reply: We have added these very important literatures to our revised manuscript.

9. Yang, L. *et al.* Ligand-directed conformational control over porphyrinic zirconium metal–organic frameworks for size-selective catalysis. *J. Am. Chem. Soc.* **143**, 12129–

12137 (2021).

14. Yue, Y. *et al.* Conductive metallophthalocyanine framework films with high carrier mobility as efficient chemiresistors. *Angew. Chem., Int. Ed.* **60**, 10806-10813 (2021).

In addition, we also cite one more important literature about the application of metal-organic frameworks.

10. Huang, N. *et al.* Flexible and hierarchical metal–organic framework composites for high-performance catalysis. *Angew. Chem., Int. Ed.* **57**, 8916-8920 (2018).

The relevant literatures are added on page 18 in the revised manuscript.

REVIEWERS' COMMENTS

Reviewer #1 (Remarks to the Author):

The comments raised by reviewers were well addressed in revised manuscript. I recommend the publication of revised manuscript in Nature Communications.

Reviewer #2 (Remarks to the Author):

The authors have made revisions and solved all the issues according to the reviewers' suggestions. The revised manuscript is well-organized and suitable for the wide readership of Nature Communications. I recommend its publication as it is now.

RESPONSE TO REVIEWERS' COMMENTS

To Reviewer 1:

General comment: The comments raised by reviewers were well addressed in revised manuscript. I recommend the publication of revised manuscript in Nature Communications.

Reply: Thank you very much for your positive comments and recommendation on the manuscript.

To Reviewer 2:

General comment: The authors have made revisions and solved all the issues according to the reviewers' suggestions. The revised manuscript is well-organized and suitable for the wide readership of Nature Communications. I recommend its publication as it is now.

Reply: We wish to express our sincere thanks to the referee for the encouraging evaluation.